# The Characterization of Ukrainian Volcanic Tuffs from the Khmelnytsky Region with the Theoretical Analysis of Their Application in Construction and Environmental Technologies

**DOI:** 10.3390/ma14247723

**Published:** 2021-12-14

**Authors:** Yuliia Trach, Victor Melnychuk, Magdalena Maria Michel, Lidia Reczek, Tadeusz Siwiec, Roman Trach

**Affiliations:** 1Institute of Civil Engineering, Warsaw University of Life Sciences–SGGW, 02-787 Warsaw, Poland; yuliia_trach@sggw.edu.pl (Y.T.); roman_trach@sggw.edu.pl (R.T.); 2Department of Water Supply, Water Disposal and Drilling Engineering, National University of Water and Environmental Engineering, 33028 Rivne, Ukraine; v.g.melnychuk@nuwm.edu.ua; 3Institute of Environmental Engineering, Warsaw University of Life Sciences–SGGW, 02-787 Warsaw, Poland; lidia_reczek@sggw.edu.pl; 4Department of Environmental Engineering and Geodesy, University of Life Sciences in Lublin, 20-069 Lublin, Poland; tadeusz.siwiec@up.lublin.pl

**Keywords:** mineral resources, natural materials, properties recognition, sorption and redox properties

## Abstract

(1) The mineral deposits are the base resources of materials used in building and environmental engineering applications, especially available locally. Two wells of volcanic tuff deposits in the Khmelnytsky region of Ukraine were investigated in this regard. (2) Physical-mechanical, chemical, and mineralogical analyses of the core samples were carried out. (3) The tuff samples were characterized by visible colour, low compressive strength (4.34–11.13 MPa), and high water absorption (30%). The dominant minerals of the upper horizon were chlorite, pyroxene, kaolinite, quartz, hematite, and calcite, while those of the lower horizon included analcime, quartz, hematite, and calcite. (4) The studied volcanic tuffs seem to be only partly useful for construction applications, and considering their visible colour, the exterior decoration of engineering objects could be possible. The peculiarity of the minerals of the upper horizon is that their crystals consist of Fe^2+^. An analysis of existing scientific data made it possible to say that these minerals can be considered as an alternative to expensive metallic iron in reducing the toxicity of chromium, uranium, and halogenated organic compounds. The significant presence of hematite allows the application of tuffs to technologies of water purification from As^5+^, As^3+^, Cr^6+^, Cr^3+^, U^6+^, Sb^5+^, and Se^4+^ oxyanions.

## 1. Introduction

Volcanic tuffs are natural building materials that have been used since historical times because of their softness, ease of processing, and low thermal conductivity [1,2]. Their popularity follows from their common occurrence and a large variety of colours, textures, compositions, and grain sizes [3].

Volcanic tuffs, the natural pozzolans, are the supplementary cementitious materials applied in the environment-friendly concrete industry [4]. Depending on the characteristics, volcanic tuffs are considered as replacements for Portland cement at different replacement levels. As it has been shown, the addition of volcanic tuff can increase the compressive and flexural strength of mortar [5,6] as well as improve the resistance of concrete to sulphates and alkali-silica reactions [7]. Zeolite-rich tuffs, which contain a large amount of reactive SiO_2_ and Al_2_O_3_, are able to increase the flexural and compressive strength of concrete blocks in the late hardening phase [8]. According to the paper of Ababneh and Matalhah [9], the quantities of volcanic tuffs available in Jordan are high and estimated at 800 million tons. The chemical composition rich in SiO_2_, Al_2_O_3_, and Fe_2_O_3_ qualifies these materials as supplementary cementitious materials. Jordanian volcanic tuffs were classified as natural pozzolans for use in concrete production. At a 10% replacement level, the volcanic tuffs produced compressive strengths comparable to those obtained when Portland cement was used alone [9]. Moreover, the alkali-silica expansion was reduced when the replacement level increased. The SiO_2_ content is positively correlated with the pozzolan activity of the additives, which was confirmed in the research on cement prepared based on Turkish tuff [10].

Volcanic tuffs are also used in road construction. The pyroclastic volcanic aggregates, due to high porosity and ability to fragmentation, can be applied for use as permeable subbases and subgrades [11]. Jordanian volcanic tuff was investigated as an addition to pavement construction, and the results showed the improvement of material properties, like strength, compaction, weathering, soundness, and resistance to abrasion [12]. In addition, zeolitic tuff can be considered as a partial substitute for lime filler in the warm asphalt mix technology and does not adversely affect the mastic properties of asphalt [13]. However, some investigations report that using the Abakaliki pyroclastic rock aggregates for road construction in a hot climate can exhibit long-term poor field performance even though the material parameters of aggregates meet the recommended limits [14].

Rock materials are characterized by remarkably diverse properties, but the adsorption properties are especially significant in the removal of heavy metals and organic compounds [15,16,17]. Locally available mined rocks are often tested for use in removing pollutants from water or wastewater [18,19,20,21,22,23]. The practical use of rocks for environmental engineering applications is determined by their hardness, mechanical resistance, density, grain morphology, and structural properties but mainly by the mineral composition. Mineral materials widely used in environmental technologies are clay materials and zeolites. Bentonite clay consists of smectite, which is a commonly used material in geosynthetic clay liner [24], a passive technique of prevention the environment from pollution. Another solutions is the filled permeable reactive barrier, e.g., zeolite, which is proposed for protecting the environment against leachate from old landfills or runoff water from urbanized areas [25,26].

The clay minerals are applied for the removal of heavy metals from water due to their packet structure and the presence of exchangeable cations [27]. Furthermore, the modified clays are effective adsorbents of inorganic anions, pharmaceuticals, and herbicides and are materials of bactericidal properties [28]. The often-used natural origin mineral ion exchangers are zeolites and glauconites. The crystal lattice of both aluminosilicates and phyllosilicates contains excessive negative charge compensated by exchangeable cations. The cation exchange capacity of zeolites and glauconites may vary by origin and is equal to 64–229 meq/100 g [29] and 11–35 meq/100 g [30,31,32], respectively. As a result, zeolites and glauconites can be used as adsorbents of heavy metals and radionuclides [29,30,33,34,35]. A feature of glauconite often used in water treatment by filtration in active techniques is the occurrence in the form of granular glauconite sands.

Volcanic tuffs are a type of pyroclastic rock formed from a material that is released during a volcanic eruption. These are fallout or flow deposits consisting of ash and dust compacted and cemented into rock. Tufts have diverse mineralogy [36,37] since it depends on the type of magma (basaltic, andesitic, rhyolitic). Additionally, pyroclasts can transform into zeolite, montmorillonite, or kaolinite during diagenetic processes, which further increases the diversity of their mineral composition [38]. The tuffs also contain various cementing components, silica and carbonate minerals, clay minerals, and iron oxides [38]. The mineral diversity of tuffs is of decisive importance in terms of their engineering applications.

Zeolitic tuff (from the region of Barsana, Maramures, Romania) containing clinoptillolite (zeolite), smectites (clay minerals), and seladonite (mica group) was able to remove Pb^2+^ and Zn^2+^ by ion exchange [39]. What is worth emphasizing is that the research was conducted with the use of wastewater from a slag granulation basin in a metallurgical factory. Volcanic tuff rich in zeolite also has been assessed as a natural material helpful in removing Pb^2+^ from water in a slightly acidic environment [40]. Other studies have confirmed that zeolitic tuff is a potential low-cost adsorbent for removal of Cr^6+^, Fe^2+^, Cu^2+^, Zn^2+^, and Pb^2+^ from pharmaceutical wastewater [41]. The Jordanian volcanic tuff, rich in phillipsite (zeolite) and hematite (ferric oxide), effectively removed phosphates from the water [42]. The literature reports also the results of research on Ukrainian volcanic tuff mainly consisted of saponite (clay material), an adsorbent of heavy metals, such as Mn^2+^, Ni^2+^, and Pb^2+^ [43,44]. The adsorption occurs quickly in a slightly acidic environment and at a low temperature of 10 °C, the typical conditions for groundwater treatment. Additionally, the adsorption capacity of Ukrainian tuff from the Ivanodolynsky quarry (Rivne region) is twice that of basalt and zeolite [44,45].

The territory of the western part of Ukraine is rich in deposits of volcanic tuffs, which make up a huge Babin formation of the Volyn series of the Lower Vendian. They make up a whole province of aluminosilicate and other raw materials with possible unique adsorption or ion exchange properties. In the territory of Ukraine, tuffs form a volcanic-clastic cover with an area of approximately 200,000 km^2^ and an average thickness of 100 m [13]. Their estimated resources are over billion tons, which shows the importance of tuffs as a potential raw material for the region. This prompts to conduct exploratory work within the deposits and search for potential applications of Ukrainian tuffs, e.g., in the field of construction and environmental technologies.

The most well-known and studied are the Ukrainian Tashkivske and Varvarivske volcanic tuff deposits, which have reserves of about 60 million tons [13]. The correlation between the chemical and mineralogical compositions of volcanic tuffs from Varvarivske quarry and their depth position was analysed. It was established that the dominant mineral is saponite, the trioctahedral smectite. The saponite stratum of the Varvarivske deposit is two-layered: interval 20.3–35.8 m of saponite horizon (50–70% of saponite); 35.8–68.9 m of analcime-saponite horizon (saponite 40–60% and analcime 20–35%) [46]. According to chemical analysis, these deposits contain a significant amount of petrogenic Fe_2_O_3_ (12.30–12.72%) [46]. The mineralogical composition of a volcanic tuff sample taken from a depth of 47.5 m from Ivanodolynsky quarry (Rivne region, Ukraine) was mainly the saponite at 56%, quartz at 22%, and Fe_2_O_3_ at 17% [44]. Tsymbalyuk [47] reported the chemical composition of basalt tuff taken from the Politske–2 quarry (Rivne region, Ukraine) as follows: SiO_2_, 67.44%; Al_2_O_3_, 12.82%; Fe_2_O_3_, 10.14%; and TiO_2_, 1.75%. Unfortunately, there are no data concerning the stratigraphic position of sample occurrence and its mineralogical composition.

Geologic and geochemical information of volcanic tuffs and associated minerals from the Rivne region in Ukraine is essential to determine those characteristics that affect the use in environmental and technical applications. Industrial use of tuffs from different locations will vary due to differences in physical and chemical properties depending on the origin. The aim of this work is to characterize the physical, mechanical, chemical, and mineralogical properties of volcanic tuffs from two wells located in the Khmelnytsky region (Ukraine). Based on these results and literature reports, a theoretical analysis of tuff use in engineering applications, such as construction and environmental technologies, was carried out. The purpose of these considerations is to guide further research on the economic use of Ukraine’s resources.

## 2. Materials and Methods

### 2.1. Samples Origin, Sampling and Preparation

The objects of the studies were volcanic tuffs from two boreholes in the village Radoshivka of Khmelnytsky region, Ukraine. The location of the Radoshivka area in the tuff occurrence belt in northwest Ukraine is shown in Figure 1 (number 7). A more precise location is shown in Figure 2. The wells were named Radoshivka–1 (R–1) and Radoshivka–2 (R–2). In this study, two sediment cores identified at 18.0–63.5 m depth (R–1) and 19.6–86.5 m depth (R–2) were examined, of which four and five samples were selected, respectively. A detailed description of the samples is presented in Table 1.

The core samples were obtained using a crown drill with a diameter of 76 mm. The samples were selected from layers of tuff that differed significantly in visual characteristics. Core lengths of 0.3 m were taken, and the collected samples were stored in airtight containers. Preparation of samples for chemical and mineralogical analysis was performed according to the standard schemes, which includes drying, grinding in the mill, and the tuff flour quartering to reduce the sample volume.

### 2.2. Physical-Mechanical Analyses of Tuff Samples

The core samples R–1 and R–2 were tested in terms of physical-mechanical properties according to the following parameters: moisture content, density, compressive strength, and water absorption. The obtained data are the average values of the measurements performed in duplicate, the difference of which did not exceed 5%.

The moisture content (*w*) was determined by the oven-dry method at the temperature of 105 °C for 24 h and was calculated as follows [48]:(1)$$w=\frac{m_{m}-m_{d}}{m_{d}}\;\cdot \;100\%,$$
where *m_m_* (g) is weight of moist sample and *m_d_* (g) of dry sample. The rock sample weight was 200 g and the balance with an accuracy of 0.01 g was used. The procedure was terminated when the difference in mass in consecutive control weighing was less than 0.01 g.

The bulk density of tuffs was determined using core cutter method. The bulk density of naturally moist samples (*ρ*) was calculated [48]:(2)$$\rho =\frac{m_{m}}{V},$$
while the bulk density of samples (*ρ_d_*) dried at 105 °C was calculated as [48]:(3)$$\rho_{d}=\frac{m_{d}}{V},$$
where *V* (cm^3^) is the volume of tuff sample in natural state.

The compressive strength (*R*) was measured using rock samples cut from the cores and having the shape of cubes with a side length of 2 cm [49]. For the uniaxial compressive strength test, the hydraulic press Unitronic 200 kN (Matest, Arcore, Italy) was used. A testing speed of 6 mm/h was applied.

The water absorption capacity (*WA*), understood as the ability of a rock to absorb and retain water, was determined using percolation method. Water absorption was expressed by weight (*WA_w_*) as the ratio of the weight of water absorbed (*m_w_*) to the initial weight of the rock sample (*m_d_*) (% *w*/*w*) [50]:(4)$$WA_{w}=\frac{m_{w}}{m_{d}}\;\cdot \;100\%,$$

The water absorption expressed by volume (*WA_v_*) was calculated as follows [50]:(5)$$WA_{v}=\frac{V_{w}}{V}\;\cdot \;100\%,$$
where *V_w_* (cm^3^) is volume of the water retained by sample.

### 2.3. Chemical and Mineralogical Analyses of Volcanic Tuff Samples

The quantitative chemical analysis of the studied tuff samples was determined by an *X*-ray spectral analysis method using ARL Advant’X IntelliPower 1200 apparatus (Thermo Fisher Scientific, Waltham, MA, USA). Lithium metaborate–lithium tetraborate flux, containing lithium nitrate as an oxidizing agent, was fused with the sample and then placed in a platinum mould (Figure 3). The resultant disk was in turn analysed by XRF spectrometry combined with a loss-on-ignition at 950 °C. To determine the final result, both types of data were used.

Mineral phase analysis of tuff samples was carried out using Aeris Minerals PANanalytical diffractometer (Malvern Panalytical, Malvern, UK). Step scan XRD data (8.01–64.965° 2θ, 0.022° 2θ step width, 2.1 step/s) were collected for bulk samples applying Cu–Kα (λ = 0.15418 nm) radiation. The software Match! (version 3, Crystal Impact GbR, Bonn, Germany) was used to process the obtained data and determine the mineral composition of volcanic tuff samples. The phase content was quantified by the Rietveld technique. The quality of the refinement was determined by the numerical indicators *R_p_* and *R_wp_*, being below 10%. To determine the phases, the diffraction pattern of the analysed sample was compared with a reference database COD–Inorg REV218120.

## 3. Results

The investigated physical and mechanical parameters suggest that the differences between the petrographic varieties of tuffs are insignificant. The results of physical and mechanical analyses of the studied volcanic tuffs are presented in Table 2. The maximum moisture content for core R–1 for point R–1(47.2) is 7.4%, and the minimum is 4.2% for point R–1 (18.0). The maximum moisture content for core R–2 for point R–2(60.8) is 9.9%, and the minimum value is 3.3% for point R–2(86.2). The obtained range of average values of natural density was 1.92–2.66 g/cm^3^ for all tuff samples, and the range of dry density values was 1.79–2.45 g/cm^3^. The numerical values of these physical and mechanical characteristics of tuffs indicate that R–1(18.0), R–1(27.6), and R–2(19.6) have lower values than R–1(47.2), R–2(46.6), R–2(60.8), R–2(76), and R–2(86.2).

Table 2 shows that there are significant differences in the uniaxial compressive strength in the studied tuff samples. The highest values of this parameter correspond to R–1(63.2), R–2(60.8), and R–2(76.0) and the lowest to R–2(19.6), R–1(18.0), R–1(27.6), and R–2(86.2). This phenomenon can be explained by the mineral changes taking place as these rocks are formed. The content of water or porosity decreases in the rocks as a result of the changes typical for minerals in the deep. The analysis of the test results allows to draw a conclusion that it is not only the quantity but also the type of silica mineral phases that affects the geotechnical parameters of the rocks studied.

The composition of the main elements, determined by XRF analysis of the selected samples, is shown in Table 3 and Table 4 and in Figure 4. Chemical analyses of volcanic tuff have shown that oxygen, silicon, aluminium, iron, and magnesium are the base elements of the raw samples. All samples show that the ranges of the percentage composition of the main oxides have the following values: SiO_2_, 41.65–53.27%; Al_2_O_3_, 11.98–13.60%; Fe_2_O_3_, 11.09–14.90%; and MgO, 3.06–9.86%. In terms of chemical composition, the samples of the studied volcanic tuffs are close to a number of volcanic rocks of the Tashkovske and Varvarivske deposits [46].

The performed analyses of the XRD allowed us to obtain diffractograms of all samples of the studied volcanic tuffs, which are shown in Figure 5 and Figure 6, and all identified minerals are compiled in Table 5 and Table 6. The analysis of the obtained diffractograms showed that the studied volcanic tuffs consist of several phases. It can be seen from the obtained diffraction patterns that the intensity of the reflections of each mineral phase of the tuff sample depended on its amount in the sample under study and the degree of crystallization. The degree of crystallization of the minerals was estimated by analysing the obtained reflections for each sample. That is, the higher the diffraction peak and, at the same time, the narrower the half-width, the fewer or no amorphous minerals were present in the tuff samples. All diffraction patterns show that the crystallinity of the samples is variable, and the smallest can be found in those from the surface.

Based on the results of the analysis of the chemical and mineral composition, two horizons were distinguished in the studied volcanic tuff deposits. The visualization is presented in Figure 7. The upper horizon is represented by the core samples R–1(18.0), R–1(27.6), and R–2(19.6). This horizon of volcanic tuffs is polyphasic and contains seven minerals, the percentage of which is greater than 2%. Chlorite was identified as the predominant mineral (30–35%). This was also evidenced by the high content of MgO in the samples. The accompanying clay mineral was kaolinite. In addition, core samples of the upper horizon contained pyroxene, quartz, hematite, and small amounts of calcite and anatase. The lower horizon of the tuffs can be assigned to the following core samples: R–1(47.2), R–1(63.2), R–2(46.6), R–2(60.8), and R–2(76.0). Intense *X*-ray reflection in the range from 26–28° in 2θ corresponded to the presence of analcime, which belongs to the group of zeolites. It was the main mineral present in about 40–62%. Other identified minerals are quartz, hematite, calcite, and residual anatase. Sample R–2(86.2) had the strongest *X*-ray reflection (at about 27° in 2θ). This indicated that the dominant mineral phase was quartz (about 70%). The composition of this sample was so different that it was not included in the lower horizon of the tuffs.

## 4. Discussion

### 4.1. Potential Use of Tuffs in the Building Industry

The Ukrainian volcanic tuffs are semi-consolidated pyroclastic rocks containing fine ash clasts [13,46]. Volcanic tuffs raised from wells within the Radoshivka section in the air-dry state are semi-consolidated with a low water-saturation degree. These materials do not show the ability to withstand compressive loads, and their limit for uniaxial compression is in the range of 4.34–11.13 MPa (weak and low strength). The tuffs taken from Trabzon and Bayburt had similar compressive strengths, which varied between 6.7–11.0 MPa [10]. This difference in values of compressive strength is explained primarily by the qualitative and quantitative mineralogical composition of the investigated volcanic tuff.

Swelling and water absorption of the natural materials play a decisive role in weathering of building rocks. There are two mechanisms that can act inside and between the clay minerals: the inner-crystalline (intra-crystalline) or the osmotic (inter-crystalline) [51]. To understand these mechanisms, it is very important to measure the amount and type of clay minerals present. The 1:1 and 2:1 layered clay minerals can condition this effect through their cation-exchange capacity and the water uptake provoked by the electrolyte concentration in water that caused an osmotic swelling [51]. The swelling of the tuffs can be approximately 36% [51] or range from 28 to 58% [52]. In this regard, the tuffs from Radoshivka showed similar values—the water absorption by volume ranged from 28 to 33% (Table 2).

Nowadays, given the active development of the construction industry, there is a demand for blended cements. This is becoming more and more relevant due to the need of preserving a clean environment and energy conservation. In the future, this demand will not be met by the synthetic additives available on the market. The comparative studies of natural clinoptilolite with various analcimes of natural volcanic origin were carried out [53,54]. According to the results, analcime exhibits properties similar to clinoptilolite, which is widely used in the production of cement mixes. Therefore, the authors believe that in some cases, because of these similar properties, analcime can be an alternative to clinoptilolite.

The lower horizon of volcanic tuffs from Radoshivka identified by us contains analcime (40–62%). However, apart from this, these tuffs contain a significant amount of hematite (8.5–18.5%), which determines their brown colour (Figure 3). This characteristic of investigated tuff samples may be the main criterion for not using them as an additive to cement. However, considering their visible colour, the application for exterior decoration of engineering objects could be possible.

### 4.2. Potential Use of Tuffs in Environmental Technology

The qualitative and quantitative compositions of the studied volcanic tuffs physical-mechanical, chemical, and mineralogical analyses were carried out in order to determine their most effective segment of their use. The peculiarity of minerals in the upper horizon is that all of them (chlorite, kaolinite, pyroxene, hematite, calcite) possess sorption capacity for heavy metals [55,56,57,58,59,60]. Chlorite and pyroxene can take part in redox processes. In these processes iron can be an electron donor, which is present in a bivalent form in a crystal of chlorite and pyroxene. There are scientific studies that the surface of crystals of such minerals; that is, those containing Fe^2+^ in the crystal lattice are capable of reducing Cr^6+^, U^4+^ and halogenated organic substances [43,58,61,62,63]. As stated in these works, the main conditions for the passage of redox processes with the participation of aluminosilicates, which include Fe^2+^, are an acidic environment and anoxic water areas. The electron donor Fe^2+^ in aluminosilicate crystals is retained by polar covalent bonds. In order for this iron to become active, it is necessary to destroy these bonds. Such destruction is possible in an aqueous environment with a low pH value. It is also known that in acidic environments, to restore the above contamination by Fe^2+^, minerals such as pyrite, glauconite, and siderite are used [43,64,65].

Since volcanic tuffs are not widely used, and taking into account their large reserves and distribution, it is obvious from an environmental and economic point of view that their use in environmental technologies is expedient. In an acidic environment (pH 4–5), the reduction of Cr^6+^ to Cr^3+^, with the participation of iron in a reducing form (Fe^2+^), occurs according to the following reaction [66]:(6)$$3{\mathrm{Fe}}^{2+}+{\;\mathrm{HCrO}}_{4}^{-}+3H_{2}O\;\to 3\mathrm{Fe}{\left(\mathrm{OH}\right)}_{2}^{+}+\;\mathrm{Cr}{\left(\mathrm{OH}\right)}_{2}^{+},$$

In a neutral medium, if pH ranges between 6 and 8, then the reduction of Cr^6+^ can occur according to the following reaction [66]:(7)$$3{\mathrm{Fe}}^{2+}+{\;\mathrm{CrO}}_{4}^{2-}+8H_{2}O\;\to 3\mathrm{Fe}{\left(\mathrm{OH}\right)}_{3}+\;\mathrm{Cr}{\left(\mathrm{OH}\right)}_{3}+4H^{+},$$

After reduction of Cr^6+^ to Cr^3+^ and oxidation of Fe^2+^ to Fe^3+^ at pH 4–5, positively charged hydroxides Fe^3+^ and Cr^3+^ are formed. The surface of aluminosilicates (chlorite, pyroxene, kaolinite), which are contained in volcanic tuffs, have a negative charge. Therefore, the formed products of this redox reaction will adsorb on the surface of minerals.

Dehalogenation of organic substances (for example, chloroethene), with the participation of iron in a reducing form (Fe^2+^), can occur by the following reactions [62]:(8)$$2{\mathrm{Fe}}^{2+}\to 2{\mathrm{Fe}}^{3+}+\;2e^{-},$$
(9)$$2H_{2}O+2e^{-}\to H_{2}+\;2{\mathrm{OH}}^{-},$$
(10)$$C_{2}{\mathrm{Cl}}_{4}+H_{2}+2e^{-}\;\to C_{2}{\mathrm{HCl}}_{3}+{\mathrm{Cl}}^{-},$$

Prolonged residence of halogenated organic substances in the reactive barrier promotes the decomposition of intermediate reaction products, such as 1,2-dichloroethene and vinyl chloride, for complete decomposition into ethene and ethane [62,67]. The formed dehalogenated organic substances are not toxic and subsequently undergo biological degradation in the soil. The end product of this biodegradation is the formation of carbon dioxide. Tuffs from the upper horizon of Radoshivka wells containing Fe^2+^ can be considered as an alternative to expensive materials, such as Fe^0^ [62,68].

Analcime and hematite were identified as characteristic and dominant components of the lower horizon of volcanic tuffs from Radoshivka. Analcime is zeolite characterized by very fine channels in structure, which makes it difficult for some ions to migrate inside [69]. This causes the cation exchange capacity of synthetic analcime to be an order of magnitude smaller than synthetic zeolites Na–P, Na–Y or clinoptilolite [70]. In-depth analysis showed that the ion diffusion into the analcime framework depends not only on cation size. High polarizability allows the lead cation to migrate inside the structure of analcime despite its large size, and cation entering selectivity is Pb^2+^ > Cu^2+^ > Zn^2+^ > Ni^2+^ [71]. Other results present the following removal order Pb^2+^ > Cd^2+^ > Mn^2+^ > Fe^2+^ and explain it by the size of the diameter of the hydrated ions, and the demonstrated removal efficiency was over 92% [72]. What is especially relevant is that, at ambient temperature, removal efficiency of ammonium and nitrate from the water is relatively small, 3.8% and 15%, respectively [72,73]. Additionally, the limited effectiveness of the analcime in removing Zn^2+^ (2.5%) and Ag^+^ (10%) has been demonstrated. Obviously, an increase in temperature promotes intracrystalline diffusion and significantly improves removal even to 50% at 80 °C for ammonia and to 90% at 55 °C for nitrate.

However, when the temperature is forced to be elevated, it is disadvantageous to use the tuff rich in analcime as an ion exchanger since it causes a high energy demand of the technological line. An alternative is the treatment of the analcime with hydrochloric acid, which causes a fourfold increase in the specific surface area of the material [69]. The acid activation with subsequent geopolymerisation also doubles the ion exchange capacity and promotes the removal of the ammonium ion [74]. Unfortunately, this requires the use of chemicals. Another direction of using analcime in environmental engineering is the production of nanofiltration membranes. The thin film of the composite containing crystalline analcime and amorphous geopolymer was able to effectively reject methylene blue dye with a high permeation rate, and low costs of fabrication are anticipated [75].

After the analcime, the main mineral of the second horizon is hematite. In terms of adsorption applications, hematite is a mineral with more possibilities. The use of hematite to remove oxyanions is well documented [76]. This mineral very effectively forms surface complexes with oxyanions of As^5+^ and As^3+^ [77,78,79], Cr^6+^ and Cr^3+^ [80,81,82], U^6+^ [83], Sb^5+^ [84], and Se^4+^ [85] which is most often favoured by an acidic environment. Importantly, it has been shown that natural minerals containing hematite have a similar ability to remove arsenic as their synthetic analogues [79]. The soil materials containing hematite were recognized as potential sorbents for the purification of arsenic-contaminated water [86,87]. Removal of As^5+^ and Cr^6+^ oxyanions on hematite occurs also at pH 7–9, typical for natural water, and the obtained efficiencies were on the level of 80–60% and 85–30%, respectively [77,80]. In this context, it seems reasonable to further recognize the properties of hematite-rich tuffs from the Khmelnytsky region.

The analysis of the properties of the minerals from upper and lower horizons is necessary for developing specific recommendations that can be effective in purifying and rehabilitation technologies of the environment. Volcanic tuffs can be used in active and passive methods for water purification. Such active methods are technologies that include sorption reactors in which the contact of contaminated water with the sorbent lasts from 30 to 240 min [15,44,56]. In this way, heavy metals can be removed. The passive method is to pass water through a specially organized permeable reactive barrier (PRB). The use of volcanic tuffs as a reactive PRB material can be very effective. The underground environment is anaerobic and favours the redox reaction of Cr^6+^ with Fe^2+^.

The metallic iron (Fe^0^) is often used as a reactive material in PBRs to remove Cr^6+^, U^4+^, and halogenated organics [62,67,68]. The use of metallic iron (Fe^0^) in redox processes for many years has shown its high efficiency. Yet, its use has a number of disadvantages, including the complexity of its production and consequently its high price. There are such pollutants that can be eliminated with its help in all countries of the world. Currently, the largest producers of such metallic iron are North America (Detroit, Cleveland, Southern California) and Canada (Ontario). The use of such material has a very high energy consumption during its transportation [88]. Therefore, it is worth considering the cheap sources of Fe^2+^, which can be volcanic tuffs. The redox process oxidizes the iron to a trivalent form, which inhibits the reduction of pollutants. In order to increase the economic effect of their use, it is necessary to organize the parallel implementation of the oxidized iron reduction technology or an easy exchange of these waste tuffs. The redox-altering reagent can be sodium dithionite (Na_2_S_2_O_4_) [89]. The dithionite ion, commonly known as hydrosulphite, is a strong reductant, particularly in strongly basic solutions. Reduction reactions with the dithionite anion typically proceed in two steps: dissociation of the dithionite ion to form two sulphoxyl radical anion (SO_2_^•–^), and the reaction of these radicals with the oxidized species Fe^3+^ yields a reduced species Fe^2+^ and oxidized sulphite (SO_3_^2^^–^) or bisulphite (HSO_3_^–^).

## 5. Conclusions

In the volcanic tuffs coming from two boreholes in Khmelnytsky region (Ukraine), layers that differed significantly in visual characteristics were identified. All layers were characterized by visible brown-grey and red-brown colour, low compressive strength (4.34–11.13 MPa), and high water absorption of about 30%. The oxygen, silicon, aluminium, iron, and magnesium were the base elements of the raw samples. and the percentage composition of the main oxides was: SiO_2_, 41.65–53.27%; Al_2_O_3_, 11.98–13.60%; Fe_2_O_3_, 11.09–14.90%; and MgO, 3.06–9.86%. Well-crystallized mineral phases have been identified.

Based on the results of the analysis of the chemical and mineral composition, two polyphasic horizons were distinguished in the studied volcanic tuff deposits. The upper horizon contained chlorite (30–35%), quartz (15–20%), kaolinite (16–18%), pyroxene (10–11%), hematite (8.5–10%), calcite (8–10%), and small amounts of anatase (2–2.5%). In the lower horizon, analcime (40–62%), quartz (15–35%), hematite (15–18.5%), calcite (5–10%), and residual anatase (1.5–2%) were found.

Given the high value of water absorption value, the tuffs cannot be an effective supplementary cementitious materials (SCMs). The distinctive colour also makes it impossible to use them as an SCMs despite the significant content of analcime in the lower horizon. The colour of the tuffs suggests that they could be used as an economically viable fine aggregates for the exterior design of engineering objects.

Based on the obtained results and literature analysis, we concluded that the two potentially expedient horizons can have an engineering and economic effect on environmental technologies. The peculiarity of the upper and lower horizons is that they contain minerals that have good sorption properties to heavy metals. The special value is the upper horizon with minerals (chlorite and pyroxene) containing iron (Fe^2+^) in the crystal structure. It is a potential line of research to check whether volcanic tuffs from Khmelnytsky region (Ukraine) can be effective in technologies for reducing the toxicity of Cr^6+^, U^4+^, and halogenated organic matter. Thus, volcanic tuffs of the upper horizon can become an alternative to expensive metallic iron (Fe^0^).

The chemical and mineralogical characteristics of the second horizon made it possible to predict the sorption properties of tuffs due to the presence of analcime and hematite. Analcime appears to be a moderate ion exchanger with potential application in Pb^2+^, Cu^2+^, and Cd^2+^ ions removal. Efficient water treatment can be expected with ions with a small hydration radius. The significant content of hematite in tuffs guides the recognition of potential application for the removal of As^5+^, As^3+^, Cr^6+^, Cr^3+^, U^6+^, Sb^5+^, and Se^4+^ oxyanions from the water.

## 6. Patents

Trach Y., Melnichuk V., Michel M., Reczek L., Melnichuk G. CПOCIБ OЧИЩEHHЯ ҐPУHTOBИX BOД BIД ГAЛOГEHIЗOBAHИX OPГAHIЧHИX PEЧOBИH TA ШECTИBAЛEHTHOГO XPOMУ ЗA ДOПOMOГOЮ BУЛKAHIЧHOГO TУΦУ (Method of treatment of groundwater from halogenated organic substances and hexavalent chromium using volcanic tuff). Patent in Ukraine No. UA 147129 U, 14 April 2021.

## Figures and Tables

**Figure 1 materials-14-07723-f001:**
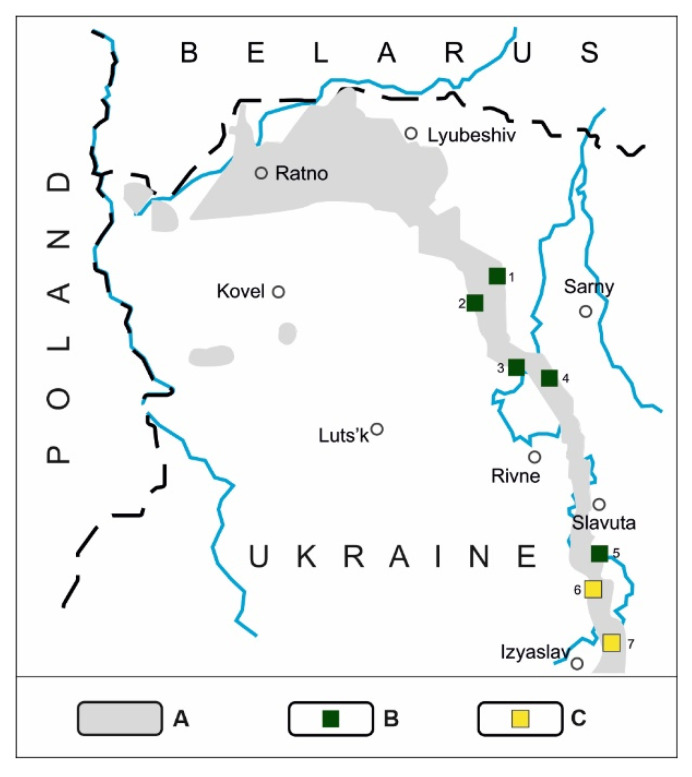
Location of volcanic tuffs of the Lower Vendian in the Volyn–Podilsky plate in Ukraine: A—protrusions of tuffs on the pre-Cretaceous surface; B—protrusions of tuffs on the surface in quarries (green squares): 1—Polytsi, 2—Rafalivka, 3—Ivanova Dolyna, 4—Berestovets, 5—Tashky; C—tuffs during recognition (yellow squares): 6—Varvarivka, 7—Radoshivka.

**Figure 2 materials-14-07723-f002:**
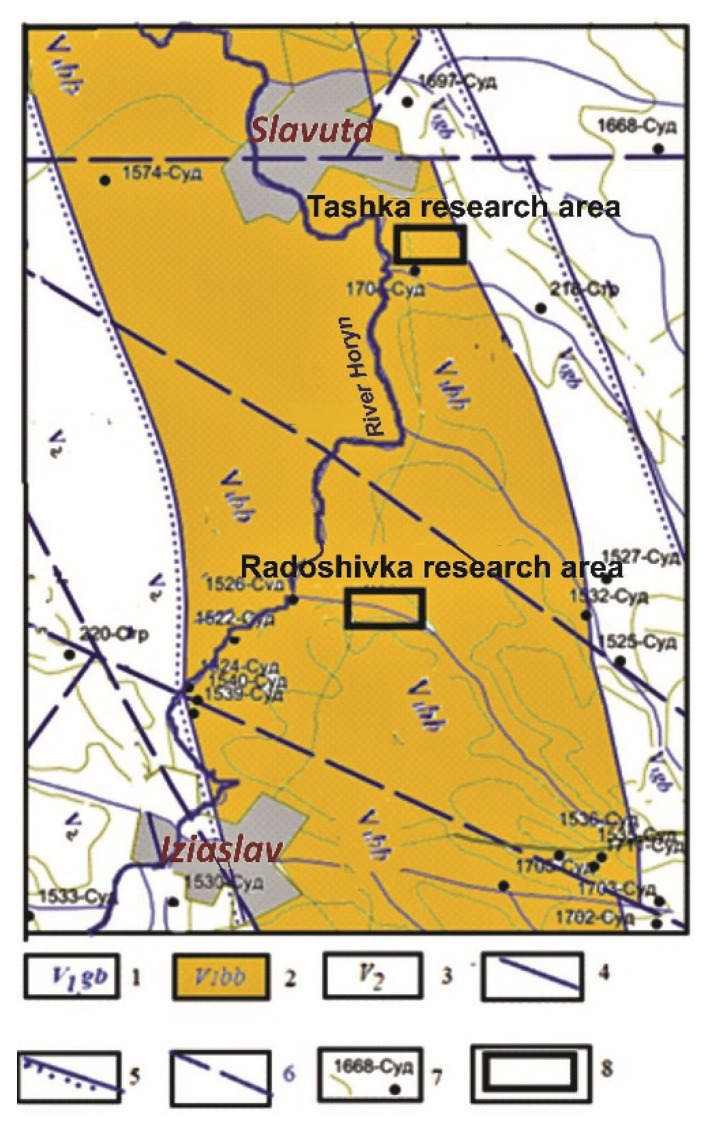
Location of the tuff stratum of the Lower Vendian in the Slavutych and Izyasldav districts of the Khmelnytskyi region: 1—strata of the Lower Vendian (Gorbashiv Formation); 2—strata of the Lower Vendian (Babin Formation); 3—strata of the Upper Vendian; 4—boundaries of strata with agreed occurrence; 5—boundaries of strata with uncoordinated occurrence; 6—probable rupture violations; 7—opened wells of the Domezozoic formations; 8—areas of detailed geological exploration of saponite tuffs in Tashka (top frame) and in Radoshivka (bottom frame).

**Figure 3 materials-14-07723-f003:**
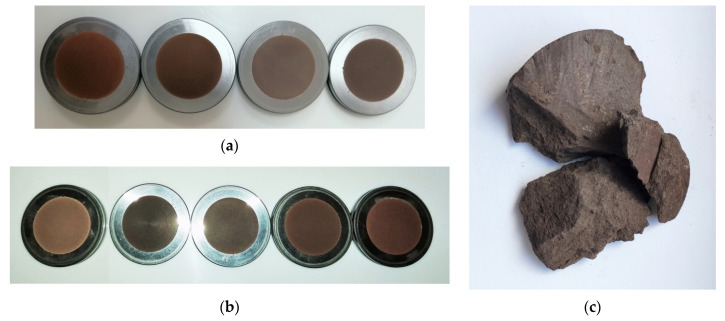
The tuff samples prepared for the XRF analysis: Radoshivka–1 well (**a**) and Radoshivka–2 well (**b**); an exemplary sample of a raw rock (**c**).

**Figure 4 materials-14-07723-f004:**
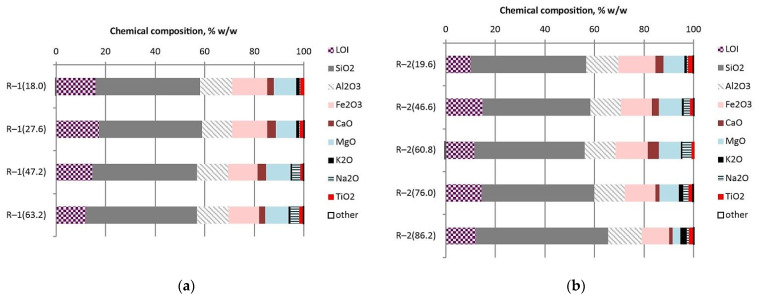
Chemical composition of volcanic tuff core samples from Radoshivka–1 well (**a**) and Radoshivka–2 well (**b**).

**Figure 5 materials-14-07723-f005:**
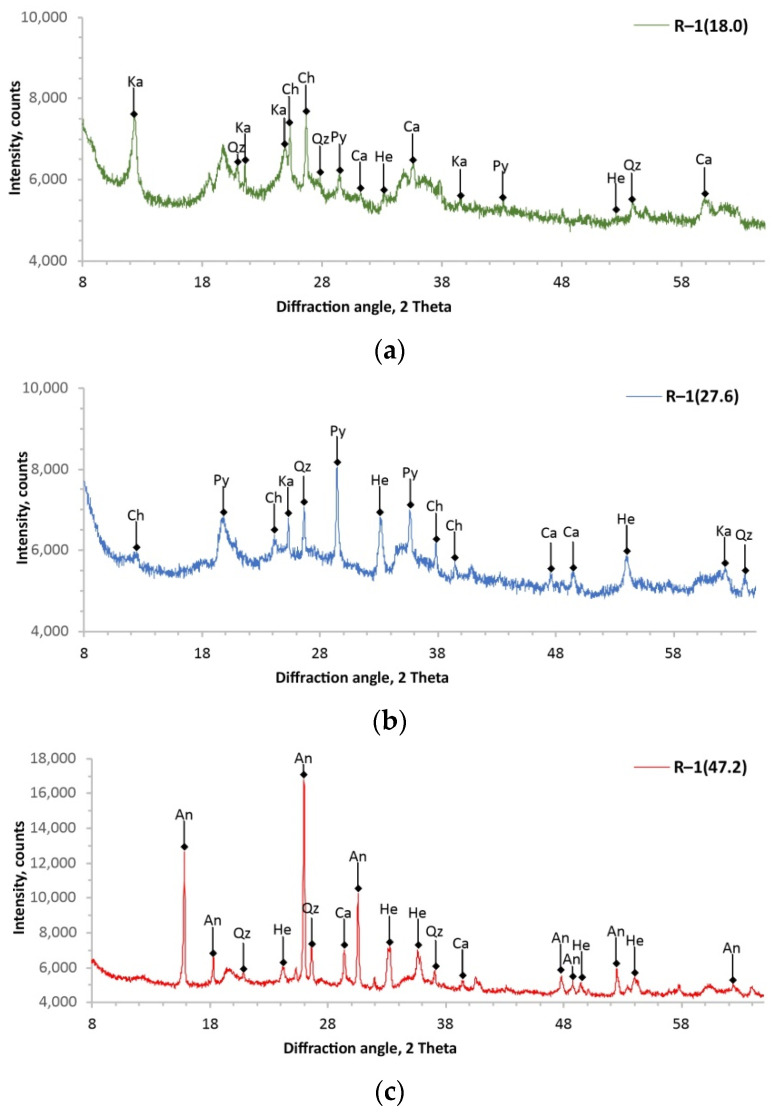
The *X*-ray patterns of core samples from Radoshivka–1 well: R–1(18.0) (**a**); R–1(27.6) (**b**); R–1(47.2) (**c**); R–1(63.2) (**d**). Ch, chlorite; An, analcime; Qz, quartz; Ka, kaolinite; Py, pyroxene; He, hematite; Ca, calcite.

**Figure 6 materials-14-07723-f006:**
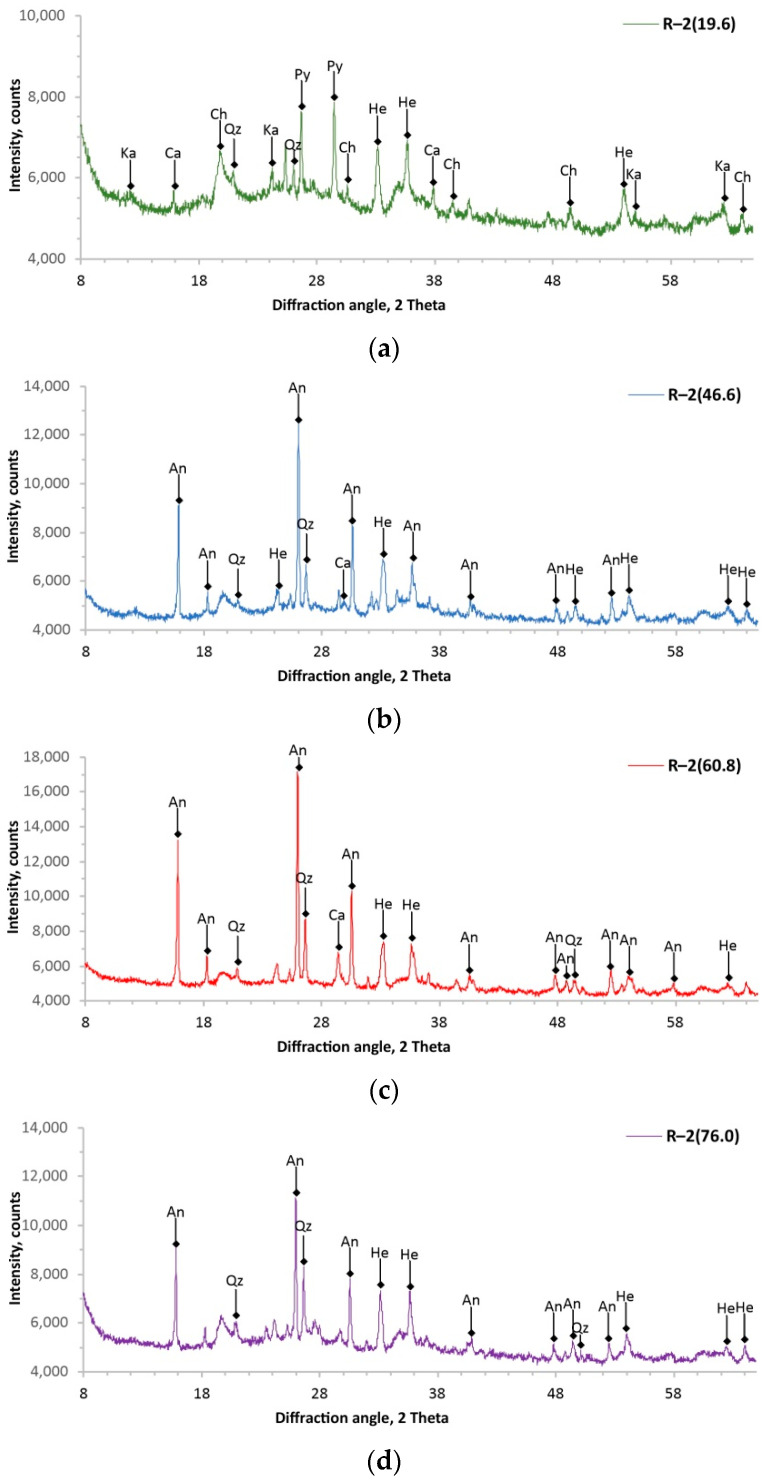
The *X*-ray patterns of core samples from Radoshivka–2 well: R–2(19.6) (**a**); R–2(46.6) (**b**); R–2(60.8) (**c**); R–2(76.0) (**d**); R–2(86.2) (**e**). Ch, chlorite; An, analcime; Qz, quartz; Ka, kaolinite; Py, pyroxene; He, hematite; Ca, calcite.

**Figure 7 materials-14-07723-f007:**
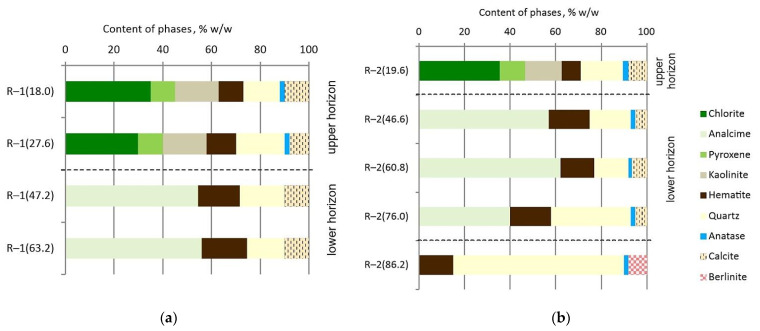
Mineral composition of volcanic tuff core samples from Radoshivka–1 well (**a**) and Radoshivka–2 well (**b**).

**Table 1 materials-14-07723-t001:** Description of the analysed samples.

Well	Depth, m	Core Sample ID	Colour Description
Radoshivka–1	18.0–18.3	R–1(18.0)	Brown-grey
	27.6–27.9	R–1(27.6)	Brown-grey
	47.2–47.5	R–1(47.2)	Red-brown
	63.2–63.5	R–1(63.2)	Red-brown
Radoshivka–2	19.6–19.9	R–2(19.6)	Brown-grey
	46.6–46.9	R–2(46.6)	Brown-grey
	60.8–61.1	R–2(60.8)	Red-brown
	76.0–76.3	R–2(76.0)	Red-brown
	86.2–86.5	R–2(86.2)	Red-brown

**Table 2 materials-14-07723-t002:** Physical and mechanical parameters of volcanic tuff samples.

Sample ID	Moisture Content, % *w*/*w*	Density, g/cm^3^	Dry Density, g/cm^3^	Compressive Strength, MPa	Water Absorption, % *w*/*w*	Water Absorption, % *v*/*v*
R–1(18.0)	4.2	2.00	1.92	5.72	18.0	33
R–1(27.6)	7.3	1.92	1.79	6.24	16.3	31
R–1(47.2)	7.4	2.46	2.29	4.63	15.8	30
R–1(63.2)	5.9	2.52	2.38	9.32	14.2	31
R–2(19.6)	4.6	2.05	1.96	7.81	17.5	33
R–2(46.6)	8.6	2.66	2.45	5.82	15.6	31
R–2(60.8)	9.9	2.45	2.23	11.13	14.1	31
R–2(76.0)	5.0	2.54	2.42	8.61	13.8	29
R–2(86.2)	3.3	2.50	2.42	4.34	13.2	28

**Table 3 materials-14-07723-t003:** Chemical composition of volcanic tuff core samples from Radoshivka–1 well (% *w*/*w*).

Components	Sample ID			
	R–1(18.0)	R–1(27.6)	R–1(47.2)	R–1(63.2)
Loss on ignition 950 °C	15.79	17.31	14.64	12.1
SiO_2_	42.23	41.65	42.27	44.76
Al_2_O_3_	12.97	11.98	12.38	12.61
Fe_2_O_3_	14.05	14.2	12.03	12.29
CaO	2.78	3.54	3.32	2.54
MgO	9.09	8.08	9.86	9.48
SO_3_	0.05	0.02	0.02	0.02
K_2_O	1.45	1.34	0.66	0.75
Na_2_O	0.13	0.19	3.22	3.62
Cr_2_O_3_	0.041	0.067	0.029	0.023
TiO_2_	1.655	1.677	1.356	1.462
Mn_2_O_3_	0.185	0.135	0.158	0.184
P_2_O_5_	0.141	0.134	0.127	0.127
SrO	0.014	0.015	0.014	0.014
ZnO	0.014	0.011	0.016	0.017

**Table 4 materials-14-07723-t004:** Chemical composition of volcanic tuff core samples from Radoshivka–2 well (% *w*/*w*).

Components	Sample ID				
	R–2(19.6)	R–2(46.6)	R–2(60.8)	R–2(76.0)	R–2(86.2)
Loss on ignition 950 °C	10.02	14.76	11.43	14.37	12.1
SiO_2_	46.6	43.58	44.62	45.51	53.27
Al_2_O_3_	13.04	12.17	12.57	12.4	13.6
Fe_2_O_3_	14.9	12.58	12.8	12.24	11.09
CaO	3.36	2.8	4.46	1.72	1.39
MgO	8.36	9.21	9.01	7.65	3.06
SO_3_	0.17	0.05	0.04	0.01	0.04
K_2_O	1.05	0.9	0.56	1.87	2.7
Na_2_O	0.39	2.41	3.57	2.1	0.9
Cr_2_O_3_	0.017	0.023	0.027	0.019	0.026
TiO_2_	1.994	1.429	1.38	1.571	1.744
Mn_2_O_3_	0.149	0.145	0.172	0.223	0.235
P_2_O_5_	0.173	0.125	0.134	0.284	0.147
SrO	0.017	0.016	0.015	0.019	0.026
ZnO	0.013	0.014	0.015	0.03	0.021

**Table 5 materials-14-07723-t005:** Mineral composition of volcanic tuff core samples from Radoshivka–1 well (% *w*/*w*).

Components	Sample ID			
	R–1(18.0)	R–1(27.6)	R–1(47.2)	R–1(63.2)
Chlorite	35	30	-	-
Analcime	-	-	54.5	56
Quartz	15	20	18.5	15.5
Kaolinite	18	18	-	-
Pyroxene	10	10	-	-
Hematite	10	12	17	18.5
Calcite	10	8	10	10
Anatase	2	2	-	-

**Table 6 materials-14-07723-t006:** Mineral composition of volcanic tuff core samples from Radoshivka–2 well (% *w*/*w*).

Components	Sample ID				
	R–2(19.6)	R–2(46.6)	R–2(60.8)	R–2(76.0)	R–2(86.2)
Chlorite	35.5	-	-	-	-
Analcime	-	57	62	40	-
Quartz	18.5	18	15	35	75
Kaolinite	16	-	-	-	-
Pyroxene	11	-	-	-	-
Hematite	8.5	18	15	18	15
Calcite	8	5	6.5	5	-
Anatase	2.5	2	1.5	2	2
Berlinite	-	-	-	-	8

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
