# Peer review of "The Characterization of Ukrainian Volcanic Tuffs from the Khmelnytsky Region with the Theoretical Analysis of Their Application in Construction and Environmental Technologies"

_materials, 2021, doi:10.3390/ma14247723_

Round 1

Reviewer 1 Report

The manuscript “Evaluation of the properties of Ukrainian volcanic tuffs from the Khmelnytsky region for applications in construction and environmental technologies” studied the physical, mechanical, chemical and mineralogical properties of volcanic tuffs from Ukraine, and their possible use in the field of constructions and environmental applications. Τhe results are well presented. The authors have employed correctly the techniques. I have found the methodological approach correct and the conclusions well explained. English is not bad and generally is easy to follow, but there are some evident grammar mistakes, which, in most cases, do not preclude the comprehension; hence the necessity of a revision, or not, would depend on the exigency of the journal in this aspect. To conclude, I suggest this manuscript to be published in the journal “Materials” after the below minor revisions:

  • The first section introduces the problem to a reader. It is well written, concise and informative. Some references should be added in the introduction for the use of different rock materials for the removal of heavy metals.

References to be cited in the introduction field:

  • Stability of calcium carbonate and magnesium carbonate in rainwater and nitric acid solutions. Energy Convers. Manag. 2006, 47, 3059–3068.
  • An Experimental Study for the Remediation of Industrial Waste Water Using a Combination of Low Cost Mineral Raw Materials. Minerals 2019, 9 (4), 207.
  • Line 115: Instead of using “The paper [40]”, it would be better to write as “Tsymbalyul [40] reports…’’.
  • Please provide the manufacturers for all the equipment used.
  • Please add all the standard test methods.

Author Response

Dear Reviewer,

Thank you very much for your excellent and constructive comments. We have modified and improved our manuscript in line with your advice to make the paper more complete and more accurate.

Below we answered your comments and parallel changes are highlighted in red in the manuscript. We also tried to improve the grammar and the changes are marked in blue.

Best regards,

Magdalena Michel on behalf of co-authors

COMMENTS and ANSWERS:

  • The first section introduces the problem to a reader. It is well written, concise and informative. Some references should be added in the introduction for the use of different rock materials for the removal of heavy metals.

References to be cited in the introduction field:

Stability of calcium carbonate and magnesium carbonate in rainwater and nitric acid solutions. Energy Convers. Manag. 2006, 47, 3059–3068.

An Experimental Study for the Remediation of Industrial Waste Water Using a Combination of Low Cost Mineral Raw Materials. Minerals 2019, 9 (4), 207.

Thank you for this tip. We cited the second very interesting article. We cannot apply the first article because it is only about a carbonate material, and carbonates have a completely different origin than volcanic rocks and also a different mechanism of pollutant removal.

  • Line 115: Instead of using “The paper [40]”, it would be better to write as “Tsymbalyul [40] reports…’’.

The change has been made to the text

  • Please provide the manufacturers for all the equipment used.

Changes have been made to the text, lines 203 and 209

  • Please add all the standard test methods.

Changes have been made to the text, references no. 48-50

Reviewer 2 Report

The manuscript provides an interesting presentation of the possible application of Ukrainian volcanic tuff based on mineralogical and chemical characterization. However, the title seems to be misleading as the method to evaluate its potential application is limited to providing sufficient evidence for such a purpose. Please consider revising the title and improving the narrative to highlight the novelty of such a study. The authors may also consider mentioning quantitatively the amount of volcanic tuff reserve in the region. Granulometry of the material of concern should have been mentioned to add value in the discussion.

Author Response

Dear Reviewer,

Thank you very much for your excellent and constructive comments. We have modified and improved our manuscript in line with your advice to make the paper more complete and more accurate.

Below we answered your comments and parallel changes are highlighted in red in the manuscript. We also tried to improve the grammar and the changes are marked in blue.

Best regards,

Magdalena Michel on behalf of co-authors

COMMENTS and ANSWERS:

The manuscript provides an interesting presentation of the possible application of Ukrainian volcanic tuff based on mineralogical and chemical characterization.

However, the title seems to be misleading as the method to evaluate its potential application is limited to providing sufficient evidence for such a purpose.

Please consider revising the title and improving the narrative to highlight the novelty of such a study.

Thank you very much for this valuable comment. This is the view that we have been missing. We propose a new title showing that this is partly a theoretical analysis

“The characterization of Ukrainian volcanic tuffs from the Khmelnytsky region with the theoretical analysis of their application in construction and environmental technologies”.

 We also use this approach for the purpose (lines 140-145):

“The aim of this work is to characterize the physical, mechanical, chemical and mineralogical properties of volcanic tuffs from two wells located in the Khmelnytsky region (Ukraine). Based on these results and literature reports, a theoretical analysis of tuff use in engineering applications such as construction and environmental technologies was carried out. The purpose of these considerations is to guide further research on the economic use of Ukraine's resources.”

 and conclusions (lines 450-456):

‘Based on the results obtained from chemical and mineralogical analyses as well as literature reports, we are predicting the use of tuffs as cheaper materials in technology for reducing the toxicity of chromium, uranium, and halogenated organic compounds in the natural environment. The chemical and mineralogical characteristics of the second horizon allow predicting the sorption properties of tuffs precisely due to the presence of analcime and hematite’.

The introduction has been reorganized, and we highlighted the specific mineral diversity of tuffs (lines 88-96):

“Volcanic tuffs are a type of pyroclastic rock formed from a material that is released during a volcanic eruption. These are fallout or flow deposits consisting of ash and dust compacted and cemented into rock. Tufts have diverse mineralogy [36,37] since it depends on the type of magma (basaltic, andesitic, rhyolitic). Additionally, pyroclasts can transform into zeolite, montmorillonite or kaolinite during diagenetic processes, which further increases the diversity of their mineral composition [38]. The tuffs also contain various cementing components, silica and carbonate minerals, clay minerals and iron oxides [38]. The mineral diversity of tuffs is of decisive importance in terms of their engineering applications.”

 and we added more information on sorption studies with the use of tuffs (lines 101-106):

“Volcanic tuff rich in zeolite also has been assessed as a natural material helpful in removing Pb2+ from water in a slightly acidic environment [40]. Other studies have confirmed that zeolitic tuff is a potential low-cost adsorbent for removal of Cr6+, Fe2+, Cu2+, Zn2+ and Pb2+ from pharmaceutical wastewater [41]. The Jordanian volcanic tuff, rich in phillipsite (zeolite) and hematite (ferric oxide), effectively removed phosphates from the water [42].”.

The authors may also consider mentioning quantitatively the amount of volcanic tuff reserve in the region.

We introduced in the manuscript the following information about the estimated amount of Ukrainian tuffs (lines 115-118):

“In the territory of Ukraine, tuffs form a volcanic-clastic cover with an area of approximately 200,000 km2 and an average thickness of 100 m [13]. Their estimated resources are over billion tons which shows the importance of tuffs as a potential raw material for the region.”.

Granulometry of the material of concern should have been mentioned to add value in the discussion.

The Ukrainian volcanic tuffs are semi-consolidated pyroclastic rocks. They are fine tuffs with clasts smaller than 0.063 mm. Therefore, no granulometric analysis was performed. We introduced this information at the beginning of paragraph 4.1.:

‘The Ukrainian volcanic tuffs are semi-consolidated pyroclastic rocks containing fine ash clasts [13,46]’.

Reviewer 3 Report

  1. The logic in Introduction should be improved. The problems and the creative points should be declared in this section.
  2. In section 4, can the authors provide some lab testing results to support the  point? It is important.
  3.  In section 5, the major results should be given. Therefore, the current conclusion should be improved. 
  4. The three pinctures attached in the appendix should be inserted into the major part. 
  5. Some photos on these minerals should be given. 

Author Response

Dear Reviewer,

Thank you very much for your excellent and constructive comments. We have modified and improved our manuscript in line with your advice to make the paper more complete and more accurate.

Below we answer your comments and parallel changes are highlighted in red in the manuscript. We also tried to improve the grammar and the changes are marked in blue.

Best regards,

Magdalena Michel on behalf of co-authors

COMMENTS and ANSWERS:

  1. The logic in Introduction should be improved. The problems and the creative points should be declared in this section.

Thank you very much for this valuable comment. The introduction has been reorganized, and we highlighted the specific mineral diversity of tuffs (lines 88-96):

“Volcanic tuffs are a type of pyroclastic rock formed from a material that is released during a volcanic eruption. These are fallout or flow deposits consisting of ash and dust compacted and cemented into rock. Tufts have diverse mineralogy [36,37], since it depends on the type of magma (basaltic, andesitic, rhyolitic). Additionally, pyroclasts can transform into zeolite, montmorillonite or kaolinite during diagenetic processes, which further increases the diversity of their mineral composition [38]. The tuffs also contain various cementing components, silica and carbonate minerals, clay minerals and iron oxides [38]. The mineral diversity of tuffs is of decisive importance in terms of their engineering applications.”

 and we added more information on sorption studies with the use of tuffs (lines 101-106):

“Volcanic tuff rich in zeolite also has been assessed as a natural material helpful in removing Pb2+ from water in a slightly acidic environment [40]. Other studies have confirmed that zeolitic tuff is a potential low-cost adsorbent for removal of Cr6+, Fe2+, Cu2+, Zn2+ and Pb2+ from pharmaceutical wastewater [41]. The Jordanian volcanic tuff, rich in phillipsite (zeolite) and hematite (ferric oxide), effectively removed phosphates from the water [42].”

and we highlighted the importance of this resource for Ukraine in terms of its availability and quantity (lines 113-120):

“They make up a whole province of aluminosilicate and other raw materials with possible unique adsorption or ion exchange properties. In the territory of Ukraine, tuffs form a volcanic-clastic cover with an area of approximately 200,000 km2 and an average thickness of 100 m [13]. Their estimated resources are over billion tons which shows the importance of tuffs as a potential raw material for the region. This prompts to conduct exploratory work within the deposits and search for potential applications of Ukrainian tuffs, e.g. in the field of construction and environmental technologies.”.

We hope our message is now more logical.

2. In section 4, can the authors provide some lab testing results to support the  point? It is important.

We theoretically evaluated the potential applications based on results of physical, mechanical, chemical and mineralogical properties (the research we have performed). Reviewer 3 is right that we should be more precise. The same view has Reviewer 2. We propose a new title showing that this is partly a theoretical analysis

“The characterization of Ukrainian volcanic tuffs from the Khmelnytsky region with the theoretical analysis of their application in construction and environmental technologies”.

We also use this approach for the purpose (lines 140-145):

“The aim of this work is to characterize the physical, mechanical, chemical and mineralogical properties of volcanic tuffs from two wells located in the Khmelnytsky region (Ukraine). Based on these results and literature reports, a theoretical analysis of tuff use in engineering applications such as construction and environmental technologies was carried out. The purpose of these considerations is to guide further research on the economic use of Ukraine's resources.”

 and conclusions (lines 450-456):

‘Based on the results obtained from chemical and mineralogical analyses as well as literature reports, we are predicting the use of tuffs as cheaper materials in technology for reducing the toxicity of chromium, uranium, and halogenated organic compounds in the natural environment. The chemical and mineralogical characteristics of the second horizon allow predicting the sorption properties of tuffs precisely due to the presence of analcime and hematite’.

3. In section 5, the major results should be given. Therefore, the current conclusion should be improved. 

We hope it has been explained in the previous point. The conclusions have been corrected.

4. The three pinctures attached in the appendix should be inserted into the major part. 

Pictures have been inserted into the main text (Figures 3-5).

5. Some photos on these minerals should be given.

The photos showing sample materials are attached in appendix A.

Round 2

Reviewer 2 Report

My comments were addressed.

Author Response

Dear Reviewer 2,

Thank you very much for your approval of our work. Your comments were valuable and helped us to improve our manuscript. Thank you for your kind cooperation.

Best regards,

Magdalena Michel on behalf of co-authors

Reviewer 3 Report

  1. The attached Fig.A1 can be inserted in the major content.
  2. The conclusion should be improved. It should declare the major findings and contributions of the study.
  3. The creative work should be highlighted.

Author Response

Dear Reviewer 3,

Thank you very much for your valuable comments. You really helped us improve our manuscript. Thank you for your kind cooperation.

Ad 1. We transferred the figure from the appendix to the main text.

Ad 2. We propose a new version of the Conclusions. The changes are highlighted in purple.

Ad 3. Unfortunately, we do not understand how to implement this remark. However, we hope the new form of the conclusions highlighted our creative work.

Best regards,

Magdalena Michel on behalf of co-authors